**ඔ | Open Peer Review** | Bacteriology | Research Article

# A clinical predictive model for pre-transplantation *Klebsiella pneumoniae* colonization and relevance for clinical outcomes in patients receiving allogeneic hematopoietic stem cell transplantation

Yu-Qi Zhang,[1,2,3] Wen-Qi Wu,[1,2,3] Jie Xu,[1,2,3] Zai-Xiang Tang,[4] Shi-Jia Li,[1,2,3] Ling Li,[1,2,3] He-Qing Wu,[1,2,3] Xiao Ma,[1,2,3] Ji-Sheng Liu,[1,5] De-Pei Wu,[1,2,3] Xiao-Jin Wu[1,2,3]

**ABSTRACT** The purpose of this study is to establish a clinical prediction model to discriminate patients at high risk of *Klebsiella pneumoniae* (KP) colonization before allogeneic hematopoietic stem cell transplantation (allo-HSCT) and evaluate the impact of KP colonization on clinical outcomes after allo-HSCT. We retrospectively collected data from 2,157 consecutive patients receiving allo-HSCT between January 2018 and March 2022. KP colonization was defined as a positive test for KP from a pharyngeal or anal swab before allo-HSCT. Logistic regression was used to build a clinical prediction model. Cox regression analyses were performed to explore the effect of KP colonization on clinical outcomes. Among all the inpatients, 166 patients had KP colonization and 581 with no positive pathogenic finding before transplantation. Seven candidate predictors were entered into the final prediction model. The prediction model had an area under the curve of 0.775 (95% CI 0.723–0.828) in the derivation cohort and 0.846 (95% CI: 0.790–0.902) in the validation cohort. Statistically significantly different incidence rates were observed among patient groups with clinically predicted low, medium, and high risk for KP infection ($P < 0.001$). The presence of KP colonization delayed platelet engraftment ($P < 0.001$) and patients with KP colonization were more likely to develop KP bloodstream infections within 100 days after allo-HSCT ($P < 0.0001$). Patients with KP colonization had higher non-relapse mortality ($P = 0.032$), worse progression-free survival ($P = 0.0027$), and worse overall survival within 100 days after allo-HSCT ($P = 0.013$). Our findings suggest that increased awareness of risks associated with pre-transplantation bacterial colonization is warranted.

**IMPORTANCE** Several studies have identified that *Klebsiella pneumoniae* (KP) is among the most common and deadly pathogens for patients in hospital intensive care units and those receiving transplantation. However, there are currently no studies that evaluate the impact of KP colonization to patients undergoing allogeneic hematopoietic stem cell transplantation. Our results confirm that pre-existing KP colonization is relatively common in a hematology transplant ward setting and negatively affects post-transplantation prognosis. Our clinical prediction model for KP colonization can support early intervention in patients at high risk to avoid subsequent bloodstream infections and improve survival outcomes. Altogether, our data suggest that increased awareness of risks associated with pre-transplantation bacterial colonization is warranted. Future studies are needed to confirm these findings and to test early intervention strategies for patients at risk of complications from KP infection.

Address correspondence to Xiao-Jin Wu, wuxiaojin@suda.edu.cn.

Yu-Qi Zhang and Wen-Qi Wu contributed equally to this article. Author order was determined on the basis of workload.

The authors declare no conflict of interest.

See the funding table on p. 11.

**KEYWORDS** *Klebsiella pneumoniae*, colonization, hematopoietic stem cell transplantation, bloodstream infections, predictive model

Infections are the most common and significant cause of mortality and morbidity in patients with hematological malignancies, especially those who undergo allogeneic hematopoietic stem cell transplantation (allo-HSCT) (1). Recently, owing to rapid changes in antimicrobial susceptibility as well as the use of new immunosuppressive agents and myeloablative conditioning regimens, the management of infectious disease risk in patients undergoing allo-HSCT is facing new challenges. For example, many patients are colonized by one or more pathogenic bacteria species, and some transplantation-conditioning drugs and antibiotics may alter the composition of the intestinal microbiome as well as damage the intestinal mucosa, putting patients at high risk for secondary bloodstream infections (BSIs) after HSCT resulting from translocation of bacteria through the injured mucosa (2, 3). These secondary BSIs are associated with high morbidity and mortality, as they can develop into severe septicemia quickly after HSCT (4–6). Also, some novel therapies, such as chimeric antigen receptor-engineered (CAR)-T cell immunotherapy, induce susceptibility to bacterial and fungal infections (7, 8). Similarly, some reports have concluded that certain treatment modalities such as plasma exchange and anti-CD20 antibodies prescribed to clear HLA antibodies prior to allo-HSCT increase the risk of infection (9, 10). However, whether these treatments affect pathogen colonization remains unknown.

Infections with Gram-negative bacteria (GNB) have been rising exponentially in recent years, and patients are at greatly increased risk of mortality from GNB infections after all-HCT (11, 12). *Klebsiella pneumoniae* (KP), a representative Gram-negative bacilli, is among the most common and deadly pathogens complicating outcomes for patients in hospital intensive care units (ICUs) and those receiving solid organ transplant or allo-HSCT (13, 14). KP colonization is particularly pervasive in patients in ICUs, and previous studies have shown that colonization by *carbapenem-resistant* KP (CRKP) can affect between 15.2% and 49% of patients in ICUs (15–17), which increases the risk of severe morbidity and mortality for these patients (18). One previous study suggested that KP is the bacteria that most frequently colonizes the pharynx of allo-HSCT recipients (19), which is consistent with other reports (20–22).

To develop a clinical predictive model of pre-existing KP colonization in patients undergoing allo-HSCT and detect the impact of KP colonization on post-transplant prognosis, we conducted a retrospective study of KP colonization status of patients before allo-HSCT.

## MATERIALS AND METHODS

### Study design and population

We performed a single center, retrospective, case-control study of patients with a KP-positive before undergoing allo-HSCT who were hospitalized at The First Affiliated Hospital of Soochow University between 1 January 2018 and 31 March 2022 to establish a clinical predictive model to discriminate patients at high risk of KP colonization. The study was approved by the Faculty Hospital Ethics Committee at the First Affiliated Hospital of Soochow University and all subjects provided written informed consent before any study procedure. The study was conducted in accordance with the Declaration of Helsinki. No potentially identifiable patient information was collected.

All inpatients received a weekly anal and pharyngeal swab while hospitalized for allo-HSCT. Cases were defined as patients with at least one positive anal or pharyngeal swab during the study period of entering the bone marrow transplant (BMT) ward and the day of stem cell transfusion. Patients who did not have any pathogen-positive swabs were analyzed as a control group. Patients with colonization by other pathogens or a clinical manifestation of infection such as pneumonia, sepsis, or fever when admitted to

the BMT ward were excluded from the study. A few patients were also excluded on the basis of important clinical information being unavailable.

## Data collection

Demographic information, clinical findings, and laboratory results were collected from medical records. Baseline data included age, sex, hematopoietic cell transplantation–comorbidity index (HCT-CI), hematological disease, days from diagnosis to allo-HSCT, details regarding administration of chemotherapy or hospitalization, status of CAR-T cell therapy before allo-HSCT, HLA antibody positivity, previous infection (such as pre-transplantation sepsis, pneumonia, perianal infection, or soft tissue infection), disease status, conditioning regimen, HSC donor type, blood type of HSC donor and recipient, dose of mononuclear cells (MNCs), and dose of CD34+ cells. In accordance with published criteria (23), the HCT-CI was determined for each patient to evaluate pre-transplant basic status. We also performed follow-up on all patients 100 days after allo-HSCT for post-transplantation infections and related complications such as BSI, cytomegalovirus (CMV) or Epstein–Barr virus (EBV) reactivation, acute graft-versus-host disease (aGVHD), hemorrhagic cystitis, *hepatic vein occlusive disease* (VOD), and thrombotic microangiopathy (TMA).

## Definitions

KP colonization was defined as a KP-positive anal or pharyngeal swab prior to the day of allo-HSCT. KP-BSI was defined a KP-positive blood culture determined by traditional blood culture sequencing or next-generation sequencing (NGS) of a blood sample from a peripheral vein or from a central venous catheter. Presence of KP-BSI was monitored from the day of conditioning until 100 days after allo-HSCT, and the onset of KP-BSI was recorded as the date of the blood sample collection. "Previous infection" was defined as the sepsis, pneumonia, perianal infection, or soft tissue infection that occurred during the hospitalization period before entering the purification ward. Overall survival (OS) was defined as the interval from allo-HSCT until death due to any cause or the last date of follow-up. Relapse was defined as blast cells ≥5% in the BM or reappearance of blast cells in the peripheral blood or any site outside the BM, including extramedullary relapse. Neutrophil recovery was defined as the first of three consecutive days with an ANC >0.5 $\times$ $10^9$ /L. Platelet recovery was defined as the first of seven consecutive days of a platelet count >20 $\times$ $10^9$ /L without transfusion support. CMV or EBV reactivation was defined as a CMV DNA viral load >100 copies/mL or an EBV DNA viral load >100 copies/m. CRKP was determined by measuring the minimum inhibitory concentration (MIC) using E-test strips (AB Biodisk, Solna, Sweden). Carbapenem resistance was defined as an ertapenem MIC ≥2 µg/mL and meropenem and/or imipenem MIC ≥4 µg/mL. All clinical events were followed until 100 days after allo-HSCT.

## Statistical analysis

For baseline clinical data, continuous variables were recorded as median and inter-quartile range and categorical variables were summarized as count and percentage. Categorical variables were analyzed using Pearson's chi-squared test or Fisher's exact test and continuous variables were analyzed using Student's *t* test. A logistic regression model was used to estimate a predictive model for KP colonization before allo-HSCT. Selection of the final prediction model was performed by backward stepwise logistic regression to determine factors to discriminate patients with KP colonization before allo-HSCT. The final model was internally validated using the bootstrap method with 1,000 repetitions. The discrimination ability of the predictive model was calculated using the area under the receiver operating characteristic curve (AUC), and the calibration power was analyzed using a calibration plot. Decision curve analysis (DCA) was used to calculate the net benefit. Continuous variables were dichotomized by maximizing Youden's index (sensitivity + specificity − 1). We calculated the score of variables that

maintained statistical significance in the multivariate regression model using the *β* coefficient, then each score from each patient was added to obtain the total score. For univariate analyses, results were reported as hazard ratios (HRs) with 95% confidence intervals (CIs), and variables with $P < 0.1$ were selected for possible inclusion in the multivariate cox regression analysis. One-hundred-day survival curves were constructed using the Kaplan–Meier method. All tests were performed with R-software version 4.2.0. All tests were two-sided, and $P < 0.05$ was considered statistically significant.

## RESULTS

### Clinical characteristics and colonization status

Between January 2018 and March 2022, 2,157 patients were admitted to the BMT ward of The First Affiliated Hospital of Soochow University for allo-HSCT. The process for screening patients is shown in Fig. 1. Among the 2,157 total patients, 166 patients (7.7%) had KP colonization and 581 (26.9%) patients had no colonizing pathogens upon testing prior to the day of the allo-HSCT. Notably, the patients with KP colonization included in the study had a KP-positive swab but no symptoms of KP infection when admitted to BMT ward. Among the 166 patients with KP colonization, 57 (34.3%) were colonized with CRKP. The baseline and clinical characteristics of two groups are listed in Table 1. All clinical events were followed for 100 days after allo-HSCT, and the median time of follow-up for survival was 408 days.

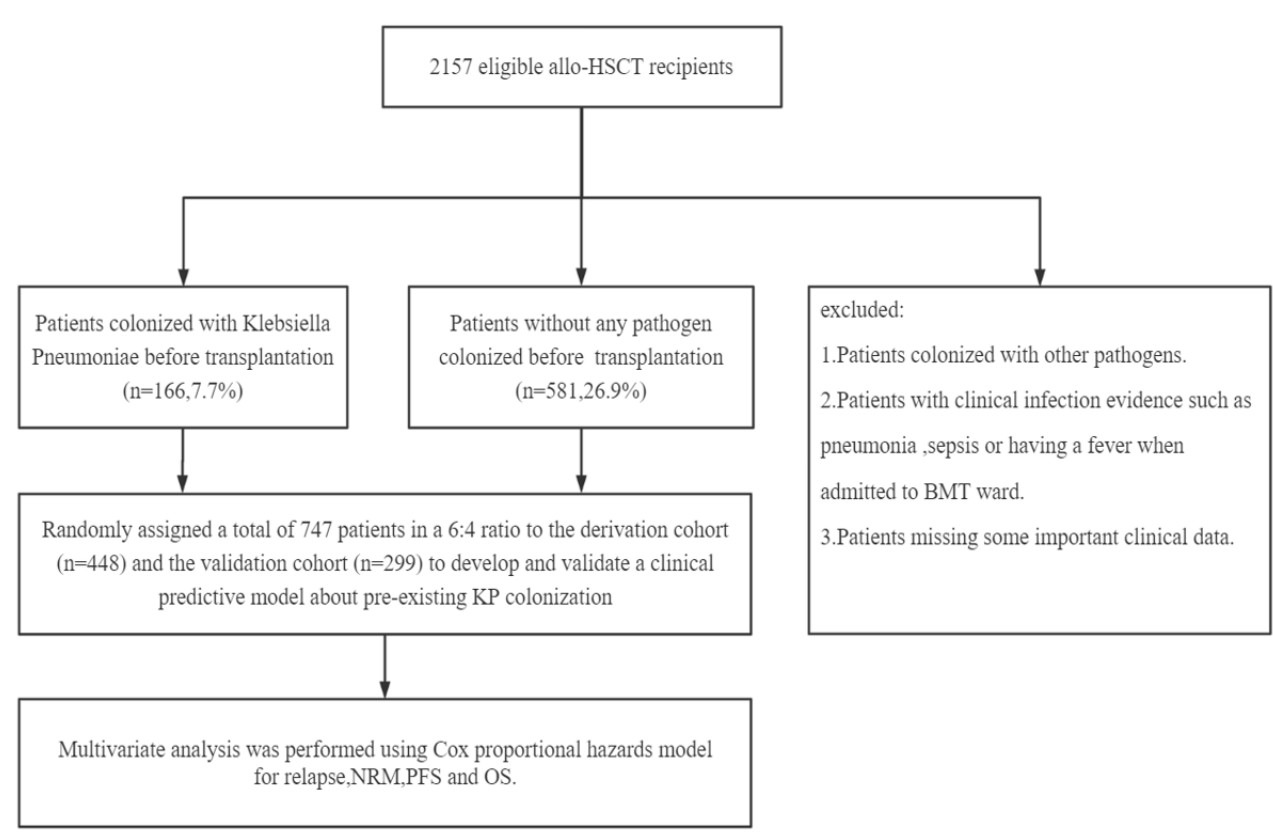

**FIG 1** Patients' enrollment and exclusion of our study.

**TABLE 1** Clinical characteristics and transplant data of KP-colonization group and non-colonization group[a]

| Characteristics | KP-colonization group (n = 166) | Non-colonization group (n = 581) | P value |
|---|---|---|---|
| Gender, no. of males (%) | 101 （60.8） | 330 （56.8） | 0.400 |
| Age, years, median （IQR） | 39 （26, 50） | 36 （27, 47） | 0.117 |
| HCT-CI | | | [b]**<0.001** |
| ≥3 | 17 （10.2） | 19 （3.3） | |
| <3 | 149 （89.8） | 562 （96.7） | |
| Underlying disease, n (%) | | | 0.069 |
| AML | 86 （51.8） | 250 （43.0） | |
| ALL | 45 （27.1） | 155 （26.7） | |
| MDS | 13 （7.8） | 78 （13.4） | |
| AA | 9 （5.4） | 24 （4.1） | |
| Others | 13 （7.8） | 78 （13.4） | |
| Days from diagnosis to HSCT, median (IQR) | 180 （139, 248） | 179 （128, 246） | 0.489 |
| Times of chemotherapy or hospitalization, median (IQR) | 3 （3, 4） | 3 （2, 4） | 0.199 |
| Car-T therapy before allo-HSCT (%) | 19 （11.4） | 49 （8.4） | 0.300 |
| HLA antigen positive (%) | 30 （18.1） | 46 （7.4） | [b]**<0.001** |
| Disease status at HSCT (%) | | | [b]**0.034** |
| CR | 129 （77.7） | 494 （85.0） | |
| Not CR | 37 （22.3） | 87 （15.0） | |
| Donor (%) | | | [b]**0.018** |
| Haploidentical | 125 （75.3） | 372 （64.0） | |
| MSD | 21 （12.7） | 123 （21.2） | |
| MUD | 20 （12.0） | 86 （14.8） | |
| Conditioning regimen (%) | | | 0.071 |
| MAC | 149 （89.8） | 547 （94.1） | |
| RIC or NMA | 17 （10.2） | 34 （5.9） | |
| Graft source (%) | | | 0.798 |
| BM | 6 （3.6） | 18 （3.1） | |
| PB | 101 （60.8） | 341 （58.7） | |
| PB + BM | 59 （35.5） | 222 （38.2） | |
| ABO incompatibility (%) | | | 0.383 |
| Compatible | 100 （60.2） | 319 （54.9） | |
| Minor mismatch | 36 （21.7） | 155 （26.7） | |
| Major/bidirectional mismatch | 30 （18.1） | 107 （18.4） | |
| Donor-recipient gender match (%) | | | 1.000 |
| Female to male | 27 （16.3） | 93 （16.0） | |
| Others | 139 （83.7） | 488 （84.0） | |
| MNC (10E8/kg), median (IQR) | 10.22 （7.32–13.84） | 8.92 （6.56–11.83） | [b]**0.001** |
| CD34+ (10E6/kg), median (IQR) | 4.07 （3.00–5.65） | 3.86 （3.00–5.13） | 0.208 |

[a]HCT-CI, hematopoietic cell transplantation–comorbidity index; AML, acute myelogenous leukemia; ALL, acute lymphoblastic leukemia; MDS, myelodysplastic; AA, aplastic anemia; CAR-T, chimeric antigen receptor-engineered (CAR)-T cell immunotherapy; MSD, matched sibling donor; MUD, matched unrelated donor; CR, complete remission; MAC, myeloablative conditioning; RIC, reduced intensity conditioning; NMA, non-myeloablative conditioning; BM, bone marrow; PB, peripheral blood; MNC, mononuclear cells.
[b]P value less than 0.05.

## Development of a KP-colonization predictive model and model performance

To develop and validate a clinical predictive model of pre-transplantation KP colonization, we randomly assigned 747 patients in a 6:4 ratio to the derivation cohort (n = 448) and to the validation cohort (n = 299). Comparison of baseline characteristics between these two cohorts revealed no statistical differences. Logistic regression was used to prioritize variables for a clinical predictive model for KP colonization, and variables for the final model were selected using backward stepwise logistic regression. The variables included in the final clinical prediction model for KP colonization were: HCT-CI score, times of chemotherapy or hospitalization, HLA antibody positivity,

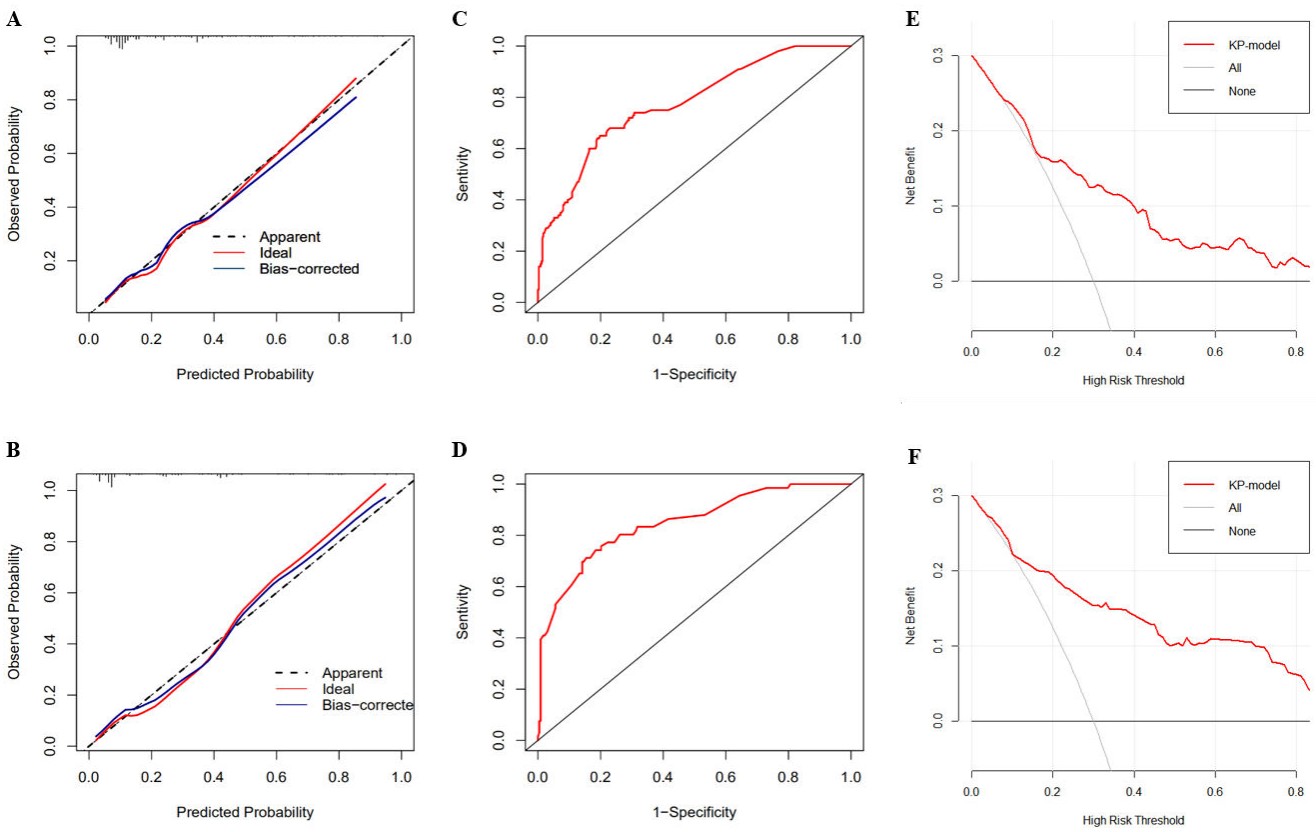

**FIG 2** The plots of pre-allo-HSCT KP colonization predictive model. Calibration plot of clinical model for predicting KP colonization of the derivation cohort (A) and the validation cohort (B). Receiver-operating characteristic curve of the derivation cohort. The AUC was 0.775 (95% CI, 0.723–0.828) (C). Receiver-operating characteristic curve of the validation cohort. The AUC was 0.846 (95% CI, 0.790–0.902) (D). Decision curve analysis of the derivation cohort (E) and validation cohort (F).

pre-transplantation pneumonia, pre-transplantation perianal infection, pre-transplantation soft tissue infection, and pre-transplantation sepsis. The percentage of times that each factor appeared in both cohorts is shown in the Table S1.

The predictive model developed in the derivation cohort had excellent discrimination, with an AUC of 0.775 (95% CI 0.723–0.828) (Fig. 2C). Additionally, the calibration plots showed good agreement between the actual probabilities and the model prediction (Fig. 2A). Internal validation also showed fair discrimination, with an AUC of 0.846 (95% CI 0.790–0.902) (Fig. 2D) and good agreement between predicted and observed cases of KP colonization (Fig. 2B). Moreover, the DCA plots show the clinical usefulness and net benefit of the clinical prediction model for KP colonization in both the derivation and validation cohorts (Fig. 2E and F). To evaluate the discrimination ability of our model, we conducted a 10-fold internal cross validation using the bootstrap method with 1,000 repetitions, which yielded an optimism-corrected C-statistic of 0.791 (95% CI, 0.672–0.911).

To incorporate the seven variables in the final clinical prediction model into a simplified prediction score, each variable was estimated using the $\beta$ coefficient obtained from the multivariable analysis (Table 2). Among 456 patients with a 0 score, 44 (9.6%) patients were positive for KP colonization before allo-HSCT compared with 85 (34.7%) of 245 patients with a score of 2–4, and 37 (80.4%) of 46 patients with a score ≥5 (Table 3). Therefore, we defined total scores of 0 points, 2–4 points, and ≥5 points as respectively

**TABLE 2** Results of the multivariable logistic regression model for the derivation cohort (*n* = 448)

| Characteristics | β | SE | OR (95% CI) | *P* value | Score |
|---|---|---|---|---|---|
| Pre-HSCT sepsis | 1.13 | 0.31 | 3.09 (1.68–5.66) | <0.001 | 2 |
| Pre-HSCT pneumonia | 1.83 | 0.21 | 6.27 (4.17–9.53) | <0.001 | 3 |
| Pre-HSCT perianal infection | 1.61 | 0.52 | 5.02 (1.80–13.92) | 0.002 | 3 |
| Pre-HSCT soft tissue infection | 1.39 | 0.68 | 4.04 (1.06–16.09) | 0.041 | 3 |
| HLA antibody positive | 1.05 | 0.30 | 2.86 (1.59–5.10) | <0.001 | 2 |
| Times of chemotherapy or hospitalization ≥ 14 | 1.52 | 0.77 | 4.57 (1.00–21.74) | 0.048 | 3 |
| HCT-CI ≥ 3 | 1.23 | 0.40 | 3.43 (1.54–7.56) | 0.002 | 2 |

the low-risk, medium-risk, and high-risk groups. The rate of pre-existing KP colonization was statistically significantly different among the three risk groups (*P* < 0.001) (Table 3).

## Clinical events and outcomes

Univariate cox regression analysis showed that the presence of KP colonization had a negative impact on the engraftment of platelets (HR = 0.7; 95% CI: 0.59–0.84; *P* < 0.001) (Fig. 3B) but no significant effect on neutrophil implantation (HR = 0.9; 95% CI: 0.76–1.08; *P* = 0.255) (Fig. 3A). We also found that patients with pre-transplantation KP colonization were more likely to develop KP-BSI than those without pre-existing KP colonization (HR = 7.7; 95% CI: 4.44–13.36, *P* < 0.001) (Fig. 3C). However, pre-existing KP colonization did not show any impact on the incidence of CMV or EBV reactivation (HR = 1.06; 95% CI: 0.79–1.02; *P* = 0.706 and HR = 0.91; 95% CI: 0.51–1.64; *P* = 0.753, respectively) (Fig. 3E and F). Among patients with pre-existing KP colonization, 60 (36.1%) patients developed aGVHD after allo-HSCT compared with 252 (43.3%) patients without pre-existing KP colonization, but this did not reach statistical significance (HR = 0.92; 95% CI: 0.7–1.2; *P* = 0.518) (Fig. 3D). The rate of severe (II–IV) aGVHD, determined using the internationally accepted Magic score for grading aGVHD, was also similar between groups (36 [18.6%] patients in the KP colonization group and 94 [16.1%] patients in the non-colonized group). Pre-existing KP colonization did not significantly improve the incidence of severe GVHD (HR = 1.18; 95% CI: 0.79–1.76; *P* = 0.407). Other clinical outcomes that were numerically higher among patients with pre-existing KP colonization compared to those without were hemorrhagic cystitis (33.1% versus 26.5%, respectively; *P* = 0.114), VOD (3.0% versus 0.8%, respectively; *P* = 0.081), and TMA (4.2% versus 2.5%, respectively; *P* = 0.402).

Time to relapse within the 100 days after allo-HSCT was similar between groups (*P* = 0.28) (Fig. 4A). However, patients with pre-existing KP colonization had higher non-relapse mortality (NRM) (*P* = 0.0031) (Fig. 4B) and worse progression- free survival (PFS) (*P* < 0.001) (Fig. 4C) within the 100-day clinical follow-up period compared to patients without pre-existing KP colonization. As is shown in Table S6, after a median follow-up of 279 days, 44 (26.5%) of 166 patients with pre-transplantation KP coloniza-tion had died, and the main cause of death was secondary infection (23/44). Among patients without pre-transplant KP colonization, after a median follow-up of 382 days, 98 (16.8%) of 581 patients had died, and the main cause of death was disease relapse (30/98). By the end of the 100-day clinical follow-up period, 16 (9.4%) patients in the group with KP colonization had died, and OS during the 100-day follow-up period was statistically significantly lower in patients with pre-existing KP colonization compared to those without (90.4% versus 94.7%, respectively; *P* = 0.013) (Fig. 4D). Univariate and multivariate Cox regression analysis showed that pre-transplantation KP colonization was an independent risk factor for NRM (HR = 2.39; 95% CI: 1.26–4.52; *P* = 0.007), PFS (HR

**TABLE 3** Recorded KP colonization rates of different risk groups in allo-HSCT patients

| Risk group | Score | Number of KP colonization in our score (%) |
|---|---|---|
| Low | 0 | 44/456 (9.6%) |
| Moderate | 2–4 | 85/245 (34.7%) |
| High | ≥5 | 37/46 (80.4%) |

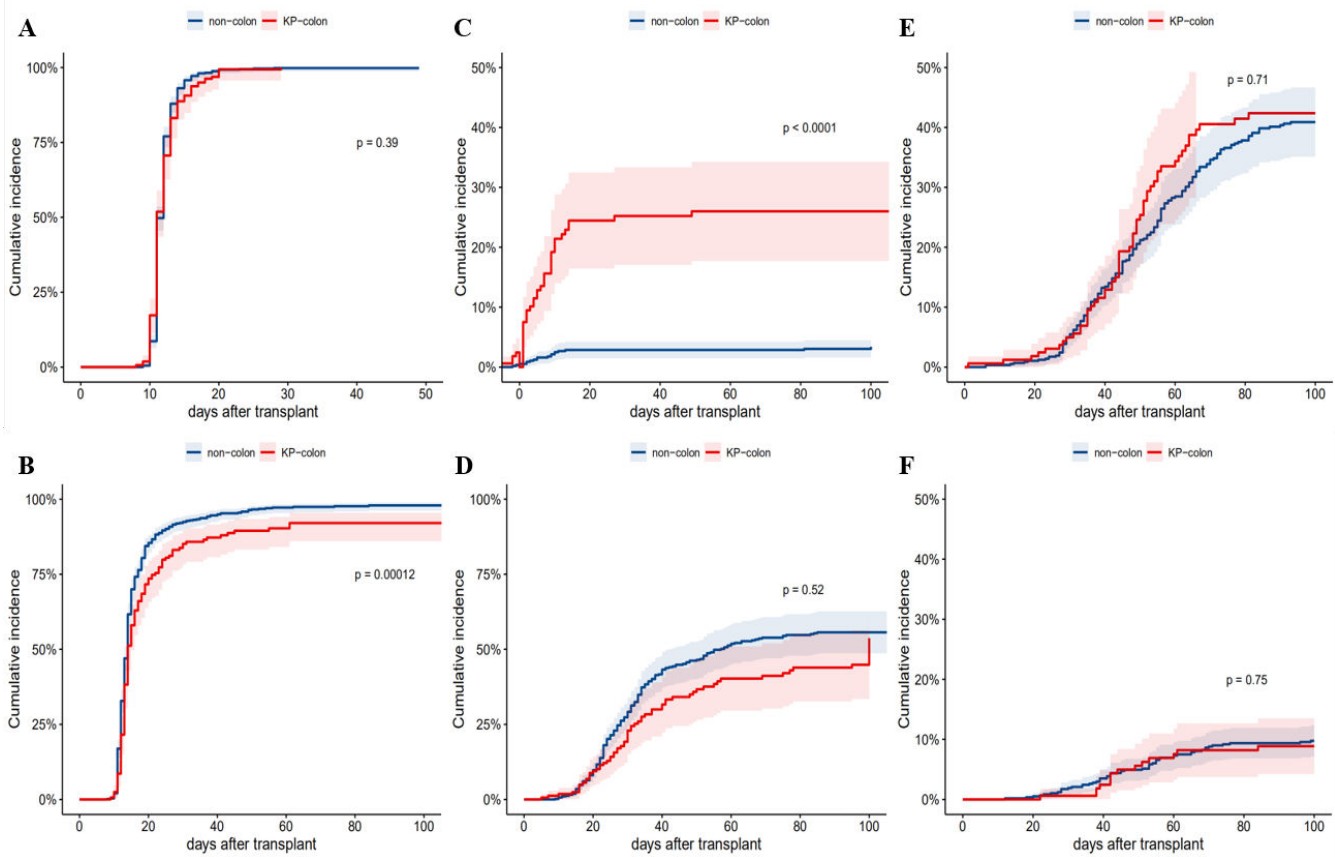

**FIG 3** Post-transplantation complications within 100 days post-allo-HSCT of KP-colonization group and non-colonization group. Time of neutrophil engraftment was not significantly longer in patients with KP colonization ($P = 0.39$) (A). Time of platelet engraftment was significantly longer in patients with KP colonization ($P < 0.001$) (B). The cumulative incidence of KP-BSI was significantly higher in patients with KP colonization ($P < 0.001$) (C). The cumulative incidence of aGVHD was not significantly higher in patients with KP colonization ($P = 0.52$) (D). The cumulative incidence of CMV reactivation was not significantly higher in patients with KP colonization ($P = 0.71$) (E). The cumulative incidence of EBV reactivation was not significantly higher in patients with KP colonization ($P = 0.75$) (F).

= 1.89, 95% CI: 1.22–2.91; $P = 0.004$), and overall survival (OS) (HR = 1.93; 95% CI: 1.04–3.60; $P = 0.038$) within 100 days of allo-HSCT among our cohort. Other factors affecting relapse, NRM, PFS, and OS are listed in the Supplementary Materials (Tables S2 to S5, respectively).

The cumulative incidence of platelet engraftment, neutrophil engraftment, KP-BSI, aGVHD, CMV reactivation, and EBV reactivation had no significant statistical difference between CRKP colonization group ($n = 57$) and carbapenem-susceptible KP (CSKP) colonization group ($n = 109$) (Fig.S1). However, compared with CSKP colonization group, patients with CRKP colonization had better 100-day PFS ($P = 0.0074$) ( Fig.S2C). The comparison of 100-day relapse, NRM and OS was listed in Fig.S2A, B, and D, respectively.

## DISCUSSION

Infections caused by KP during the transplantation period are an important cause of the high mortality rates observed early after transplantation as well as lead to longer hospital stays and greater healthcare costs. The intensive chemotherapy and/or radiation preparative regimens prescribed before HSCT can damage the mucosa and allow colonizing bacteria to enter the bloodstream, and many studies have shown that patients diagnosed with BSI after transplantation have worse OS and NRM than those without BSI (24, 25). The harsh pre-transplantation preparative regimens can also result in dysbiosis of the gut microbiome, leading to outgrowth of less favorable or pathogenic

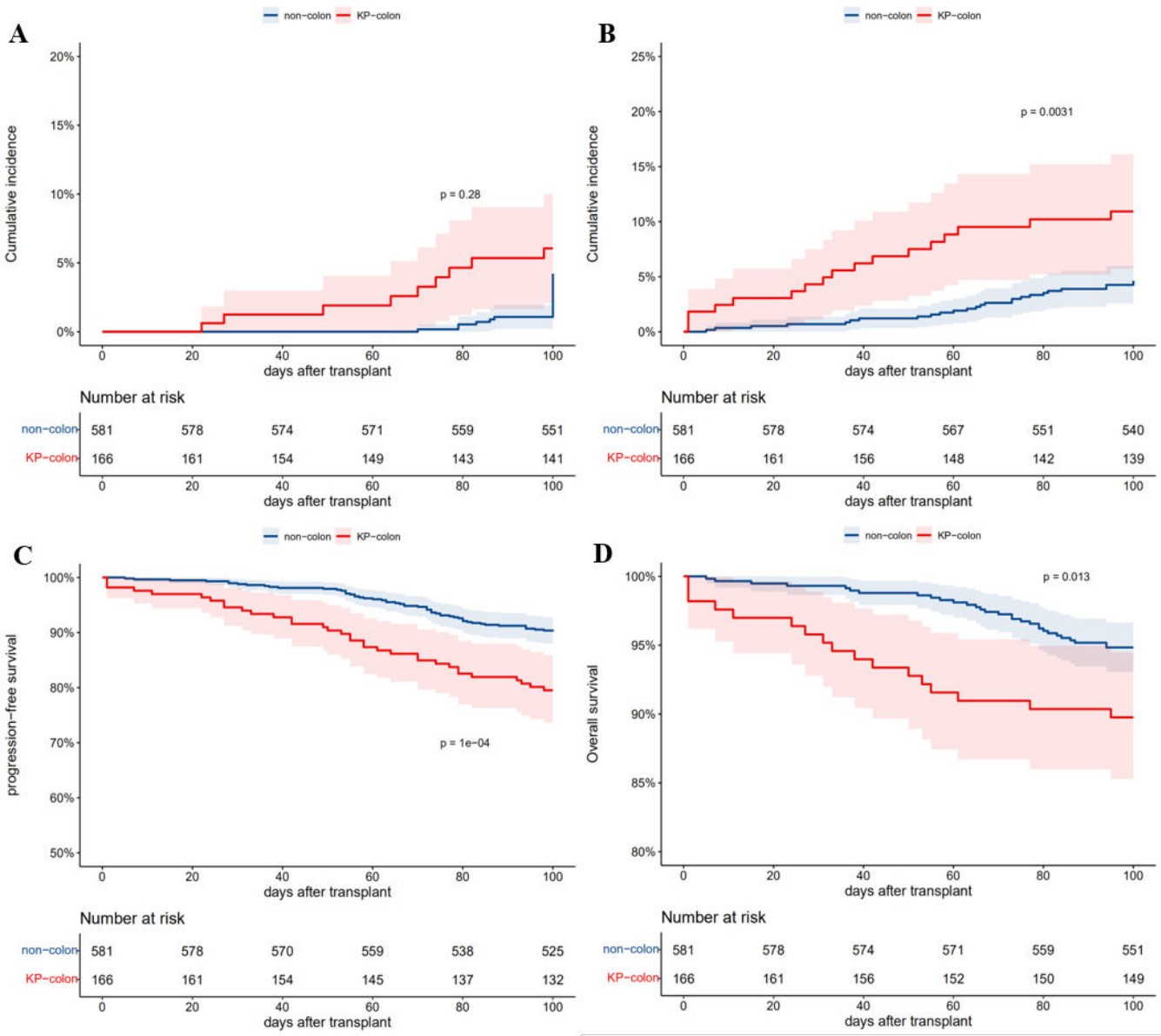

**FIG 4** Outcomes within 100 days post-allo-HSCT of recipients in KP-colonization group and non-colonization group. Relapse was not significantly higher in patients with KP colonization ($P = 0.28$) (A); NRM was significantly higher in patients with KP colonization ($P = 0.0031$) (B); PFS was significantly lower in patients with KP colonization ($P < 0.001$) (C); OS was significantly lower in patients with KP colonization ($P = 0.013$) (D).

gut microbiota, such as various species of Enterobacteriaceae (26). Indeed, KP is one of the most frequent Enterobacteriaceae species seen in hematology wards.

Previous studies have already developed several clinical predictive models about KP infection (27–29). Maddalena Giannella developed a clinical score to predict CRKP infection among inpatients with CRKP rectal carriers (27). Regarding KP, Mario Tumbarello established a clinical predictive model based on data from patients at five large Italian hospitals to predict Klebsiella pneumoniae carbapenemase-producing KP (KPCKP) isolation and KPCKP infection (28). In addition, Tumbarello found that hematological cancer is one of the independent predictors of KPCKP isolation (28). However, no predictive model about KP for allo-HSCT recipients has been developed and also none of the models was validated on pre-transplantation KP colonization. Early identification of patients at high-risk for KP colonization can be used to support early intervention such as decolonization therapy during the peri-transplantation period. "Decolonization

therapy" mentioned above including topical agents, systemic therapy, antibiotic inhaled therapy, natural compounds, bacteriophage therapy, and alternative treatments, was proved could reduce the infection risk among pathogenic bacteria carriers (30, 31). In our predictive model, the most important factors associated with pre-existing KP colonization were higher HCT-CI score, more chemotherapy treatments or hospitalizations, HLA antibody positivity, and pre-transplantation infection. The HCT-CI is used to estimate patient-specific transplant mortality risk and in our cohort, patients with higher HCT-CI scores were more likely to suffer from diabetes, liver dysfunction, and other malignant tumors, which may place them at high risk for infection and bacterial colonization. These factors can be quantified using the HCT-CI to predict which patients are more likely to be colonized by KP. More chemotherapy treatments or hospitalizations are associated with prolonged neutropenia, which can lead to infections with multi-drug resistant bacteria necessitating powerful antibiotics. One study of intestinal tract bacterial diversity concluded that treatment with intravenous vancomycin, metronidazole, or β-lactams as well as treatment with aggressive conditioning regimens for HSCT resulted in low diversity of the gut microbiome (32), which can lead to colonization by unfavorable species. Thus, strategies to identify patients at high risk of colonization by pathogenic bacteria before allo-HSCT are urgently needed.

This retrospective study reveals that patients undergoing allo-HSCT have a relatively high rate of KP colonization compared with other bacteria, which has a significant impact on prognosis early after transplantation, as well as on risk of subsequent infection and related complications. Pre-existing KP colonization was associated with increased NRM, decreased PFS, and decreased OS. Platelet reconstitution also required more time in patients with pre-existing KP colonization, whereas KP colonization status had no significant effect on granulocyte reconstitution. Recently, several studies have concluded that combining conventional tests with metagenomic NGS could significantly improve the detection rate for pathogens (33, 34). Therefore, for inpatients presenting with fever for the first time, mNGS is performed in tandem with traditional blood culture to support rapid and accurate diagnosis. Here, we found that 22.9% (38/166) of patients with pre-existing KP colonization developed KP-BSI early after allo-HSCT, which was significantly higher compared to non-colonized patients; notably, a prospective observational multicenter study in Italy and a prospective study in ICU patients drew similar conclusions (35, 36). Forcina concluded that colonization by MDR-GNB before auto- or allo-HSCT did not significantly affect OS, treatment-related mortality (TRM), or infection-related mortality (IRM) (22). Also, a study of vancomycin-resistant *Enterococcus* showed that pre-HSCT colonization was not associated with increases in HSCT-related mortality (37). However, our findings demonstrate that patients with pre-transplantation KP colonization had higher NRM as well as shorter PFS and OS, indicating that, compared with other kinds of bacteria, KP colonization and infection are highly aggressive and warrants timely and powerful antibiotic treatment. A previous study found a higher overall mortality rate associated with KP colonization and infection in patients undergoing HSCT (38), which is consistent with our research findings. Also, our multivariable analysis demonstrated that pre-existing KP colonization was an independent risk factor for NRM, PFS, and OS. Therefore, although the rate of KP colonization in patients receiving allo-HSCT was only 7.7%, considering the poor prognosis for recipients with pre-existing KP colonization, KP status should still be considered in the pre-operative period.

This study explored risk factors to discriminate patients with pre-transplantation KP colonization and established a clinical prediction model to guide clinical decision-making. It also provides the first examination of the relationship between KP colonization and post-transplantation complications. Several previous studies have shown that use of carbapenem antibiotics before chemotherapy is significantly related with subsequent colonization and infection (17). Because a considerable number of patients in our study had been treated in other hospitals before admission to our hematology ward, we could not obtain complete information on previous antibiotic use, limiting our ability to

determine whether this affected pre-transplantation KP colonization or post-transplantation outcomes. We also did not analyze the different antibiotics used to treat febrile neutropenia in this study. Finally, this study was conducted at a single hospital, and external validation of our findings is required. Because both previous studies and our data show that the colonization and infection rate for KP in hematology departments is higher than for other bacteria, our clinical prediction model may not generalize to other wards or to other pathogens.

In conclusion, our results confirm that pre-existing KP colonization is relatively common in a hematology transplant ward setting and negatively affects post-transplantation prognosis. Our clinical prediction model for KP colonization before allo-HSCT can support early intervention in patients at high risk to avoid KP-BSI and improve survival outcomes. Altogether, our data suggest that increased awareness of risks associated with pre-transplantation bacterial colonization is warranted. Future studies are needed to confirm these findings and to test early intervention strategies for patients at risk of complications from KP infection.

## ACKNOWLEDGMENTS

This work was supported by the National Natural Science Foundation of China (Grant Nos. 81974001 and 82170222); the Jiangsu Natural Science Foundation (BK20211070); The Key Disease Program of Suzhou (LCZX202101); National Science and Technology Major Project (2017ZX09304021); National Key R&D Program of China (2019YFC0840604 and 2017YFA0104502); Priority Academic Program Development of Jiangsu Higher Education Institutions (PAPD); Jiangsu Provincial Key Medical Center (YXZXA2016002); Research project of Jiangsu Provincial Health Commission (ZD2021008).

## AUTHOR AFFILIATIONS

[1]Department of Hematology, The First Affiliated Hospital of Soochow University, Suzhou, Jiangsu, China

[2]National Clinical Research Center for Hematologic Diseases, Jiangsu Institute of Hematology, Suzhou, Jiangsu, China

[3]Institute of Blood and Marrow Transplantation, Collaborative Innovation Center of Hematology, Soochow University, Suzhou, Jiangsu, China

[4]Department of Epidemiology and Statistics, School of Public Health, Faculty of Medicine, Soochow University, Suzhou, Jiangsu, China

[5]Department of Otolaryngology Head and Neck Surgery, The First Affiliated Hospital of Soochow University, Suzhou, Jiangsu, China

## AUTHOR ORCIDs

Yu-Qi Zhang http://orcid.org/0009-0001-7548-0454
Xiao-Jin Wu http://orcid.org/0000-0003-3894-0631

## FUNDING

| Funder | Grant(s) | Author(s) |
| --- | --- | --- |
| MOST \| National Natural Science Foundation of China (NSFC) | 81974001, 82170222 | Xiao-Jin Wu |
| JST \| Natural Science Foundation of Jiangsu Province (Jiangsu Natural Science Foundation) | BK20211070 | Xiao-Jin Wu |
| the Key Disease Program of Suzhou | LCZX202101 | Xiao-Jin Wu |
| National Science and Technology Major Project (国家科技重大专项) | 2017ZX09304021 | De-Pei Wu |
| MOST \| National Key Research and Development Program of China (NKPs) | 2019YFC0840604, 2017YFA0104502 | De-Pei Wu |

| Funder | Grant(s) | Author(s) |
|---|---|---|
| Jiangsu Provincial Medical Center (江苏省医疗中心) | YXZXA2016002 | De-Pei Wu |
| Research Project of Jiangsu Provincial Health Commission | ZD2021008 | Ji-Sheng Liu |

## AUTHOR CONTRIBUTIONS

Yu-Qi Zhang, Conceptualization, Data curation, Formal analysis, Methodology, Writing – original draft | Wen-Qi Wu, Data curation, Methodology, Writing – original draft | Jie Xu, Data curation | Zai-Xiang Tang, Formal analysis, Methodology | Shi-Jia Li, Data curation | Ling Li, Data curation, Validation | He-Qing Wu, Methodology | Xiao Ma, Supervision, Visualization | Ji-Sheng Liu, Funding acquisition, Project administration, Resources | De-Pei Wu, Funding acquisition, Supervision, Writing – review and editing | Xiao-Jin Wu, Funding acquisition, Investigation, Resources, Supervision, Validation

## ADDITIONAL FILES

The following material is available online.

### Supplemental Material

**Supplemental material (Spectrum02039-23-S0001.docx).** Tables S1 to S7; Fig. S1 and S2.

### Open Peer Review

**PEER REVIEW HISTORY (review-history.pdf).** An accounting of the reviewer comments and feedback.

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
