## [Reviewer comments · Microbiology Spectrum]

Microbiology Spectrum

A clinical predictive model for pre-transplantation *Klebsiella pneumoniae* colonization and relevance for clinical outcomes in patients receiving allogeneic hematopoietic stem cell transplantation

Yuqi Zhang, Wenqi Wu, Jie Xu, Zaixiang Tang, Shijia Li, Ling Li, Heqing Wu, Xiao Ma, Jisheng Liu, Depei Wu, and Xiaojin Wu

Corresponding Author(s): Xiaojin Wu, First Affiliated Hospital of Soochow University

Review Timeline:

Submission Date:	May 17, 2023
Editorial Decision:	September 4, 2023
Revision Received:	September 12, 2023
Accepted:	December 2, 2023

Editor: Fei Chen

Reviewer(s): The reviewers have opted to remain anonymous.

Transaction Report:

DOI: <https://doi.org/10.1128/spectrum.02039-23>

September 4, 2023

Dr. Xiaojin Wu
First Affiliated Hospital of Soochow University
Suzhou
China

Re: Spectrum02039-23 (A clinical predictive model for pre-transplantation *Klebsiella pneumoniae* colonization and relevance for clinical outcomes in patients receiving allogenic hematopoietic stem cell transplantation)

Dear Dr. Xiaojin Wu:

Link Not Available

Sincerely,

Fei Chen

Journals Department
Reviewer comments:

Reviewer #1 (Comments for the Author):

1. Line 94, the include variable "previous infection", how long before the KP colonization was the previous infection included?
2. Line 149 mentioned that 54 (2.6%) patients were colonized with CRKP isolates, did the author study the NRM, PFS and OS in CRKP colonized patients? Does there have any significant difference between patients colonized with CRKP an CSKP isolates?
3. Line 241-242, the author mentioned that high-risk patients could be treated with empiric antibiotics during the peri-transplantation period. Does this mean the decolonization therapy? Could the author list some support references?
4. In the abstract, the author mentioned that anal swab was taken, but in main text, the method mentioned rectal swab. So, it should be clear which method was taken.
5. Line 60, the first KP in the main text should be full name.

6. In supplementary table 2, 3, 4, why does the diagnose column different? In table 2, there were ALL and others, and in table 3 and 4, there were AA and others. And the note of AA could not be seen under the table.
7. Some formatting of punctuation marks should be corrected.

Reviewer #2 (Comments for the Author):

Comments to the Author

This study developed a clinical predictive model for pre-transplantation *Klebsiella pneumoniae* colonization and relevance for clinical outcomes in patients receiving allo-HSCT. The prediction model had an area under the curve of 0.775 and 0.846 in the derivation and the validation cohort, respectively. KP colonization negatively effects platelet engraftment and survival after allo-HSCT. My comments are as follows:

- 1) The variables for the final model were selected using backward stepwise logistic regression. Did the author try multiple algorithms and consider them comprehensively?
- 2) The author demonstrated that among the patients with KP colonization, 57(34.3%) were colonized with CRKP. Were there differences in "Clinical events and outcomes" between the KP cohort and CRKP cohort?
- 3) It is recommended that the author revise all baseline tables in the paper to three-line tables.
- 4) The caption (KP-mode, All, and None) of Figure 2 is unclear.
- 5) There are many minor spelling errors in the text. The author needs to comprehensively check and revise them through the whole manuscript.

Staff Comments:

Preparing Revision Guidelines

Please return the manuscript within 60 days; if you cannot complete the modification within this time period, please contact me. If you do not wish to modify the manuscript and prefer to submit it to another journal, please notify me of your decision immediately so that the manuscript may be formally withdrawn from consideration by Microbiology Spectrum.

The authors performed a single center, retrospective, case-control study and successfully developed a clinical prediction model for KP colonization in allo-HSCT patients. And the study found that Patients with KP colonization had higher NRM, worse PFS, and worse OS within 100 days after allo-HSCT. The study is logical and thorough, but the English language requires upgrading.

I have a few comments below for authors:

1. Line 94, the include variable “previous infection”, how long before the KP colonization was the previous infection included?
2. Line 149 mentioned that 54 (2.6%) patients were colonized with CRKP isolates, did the author study the NRM, PFS and OS in CRKP colonized patients? Does there have any significant difference between patients colonized with CRKP an CSKP isolates?
3. Line 241-242, the author mentioned that high-risk patients could be treated with empiric antibiotics during the peri-transplantation period. Does this mean the decolonization therapy? Could the author list some support references?
4. In the abstract, the author mentioned that anal swab was taken, but in main text, the method mentioned rectal swab. So, it should be clear which method was taken.
5. Line 60, the first KP in the main text should be full name.
6. In supplementary table 2, 3, 4, why does the diagnose column different? In table 2, there were ALL and others, and in table 3 and 4, there were AA and others. And the note of AA could not be seen under the table.
7. Some formatting of punctuation marks should be corrected.

Dear Editors and Reviewers:

We feel great thanks for your professional review work on our manuscript entitled 'A clinical predictive model for pre-transplantation *Klebsiella pneumoniae* colonization and relevance for clinical outcomes in patients receiving allogeneic hematopoietic stem cell transplantation' [Paper #Spectrum02039-23]. As you are concerned, there are several problems that need to be addressed. According to your suggestions, we have made extensive corrections to our previous draft. In this revised version, changes to our manuscript were all highlighted within the document by using red-colored text. Point-by-point responses to two nice reviewers are listed below this letter.

Reviewer #1:

1. Line 94, the include variable "previous infection", how long before the KP colonization was the previous infection included?

Response: Thanks for your careful comments. We have paid attention to this issue and then checked the data of these included patients. "Previous infection" defined as the sepsis, pneumonia, perianal infection or soft tissue infection occurred during the hospitalization period before entering the purification ward. After checking the clinical data, we have found that all patients' "previous infection" occurred within 6 months before transplantation. Though lacking related references, we insisted that previous infections can affect the microbiota, leading to the colonization of some pathogenic bacteria.

2. Line 149 mentioned that 54 (2.6%) patients were colonized with CRKP isolates, did the author study the NRM, PFS and OS in CRKP colonized patients? Does there have any significant difference between patients colonized with CRKP an CSKP isolates?

Response: We think this is an excellent and necessary suggestion. We have compared the clinical events and outcomes of CRKP and CSKP colonized group and concluded that, patients with CRKP colonization had worse 100-day PFS ($P=0.0074$) than those with CSKP colonization. However, 100-day relapse, NRM and OS were proved have no significant difference between two groups, with P values were 0.95, 0.07 and 0.079 respectively. We add these contents in the resubmitted manuscript and the detailed figures are shown in **Supplementary Materials**.

3. Line 241-242, the author mentioned that high-risk patients could be treated with empiric antibiotics during the peri-transplantation period. Does this mean the decolonization therapy? Could the author list some support references?

Response: Thanks for your excellent suggestions. The treatment of "empiric antibiotics" mentioned in the manuscript may not be so accurate and "decolonization therapy" may be more exact. To support this idea, we have added some references in the revised manuscript:

30. Tacconelli E, Mazzaferri F, de Smet AM, Bragantini D, Eggimann P, Huttner BD, et al. ESCMID-EUCIC clinical guidelines on decolonization of multidrug-resistant Gram-negative bacteria carriers. *Clin Microbiol Infect* 2019; **25**(7): 807-17.

31. Huang SS, Singh R, McKinnell JA, Park S, Gombosev A, Eells SJ, et al. Decolonization to Reduce Postdischarge Infection Risk among MRSA Carriers. *N Engl J Med* 2019; **380**(7): 638-50.

4. *In the abstract, the author mentioned that anal swab was taken, but in main text, the method mentioned rectal swab. So, it should be clear which method was taken.*

Response: We sincerely thank the reviewer for careful reading. As suggested by the reviewer, we have corrected the "rectal swab" into "anal swab" as in our BMT unit, all patients underwent regular anal swab examination during their hospital stay.

5. *Line 60, the first KP in the main text should be full name.*

Response: Thanks for your reminder. Based on your comments, we have made the corrections,

6. *In supplementary table 2, 3, 4, why does the diagnose column different? In table 2, there were ALL and others, and in table 3 and 4, there were AA and others. And the note of AA could not be seen under the table.*

Response: Thanks for your careful checks. We are sorry for our carelessness when uploading materials. After checking the origin tables, we have made the corrections to make the diagnosis column harmonized within the whole manuscript.

7. *Some formatting of punctuation marks should be corrected.*

Response: We feel sorry for our carelessness. We have thoroughly checked the punctuation marks of the manuscript and made the corrections.

Reviewer #2:

1) *The variables for the final model were selected using backward stepwise logistic regression. Did the author try multiple algorithms and consider them comprehensively?*

Response: We agree with you that when filtering variables, multiple algorithms and comprehensive consideration should be conducted. The methods for screening variables in logistic regression prediction model include univariate analysis, stepwise regression, LASSO and random forest. When filtering variables in our predictive model, we performed all these algorithms. However, due to the relatively few independent variables we included in our model and the weak collinearity of these included factors, we didn't take into consideration the methods of LASSO and random forest. Furthermore, similar conclusions were yielded by univariable analysis and stepwise regression (backward stepwise logistic regression and backward-forward stepwise logistic regression). Considering the independent variables, collinearity issues, and clinical experience comprehensively, we ultimately adopted backward stepwise logistic regression.

2) *The author demonstrated that among the patients with KP colonization, 57(34.3%) were colonized with CRKP. Were there differences in "Clinical events and outcomes" between the KP cohort and CRKP cohort?*

Response: We think this is an excellent and necessary suggestion. We have compared the clinical events and outcomes of CRKP and CSKP colonized group and concluded that, the clinical events such as platelet engraftment, neutrophil engraftment, KP-BSI, aGVHD, CMV reactivation and EBV reactivation had no significant statistical difference between CRKP and CSKP colonization group. However, patients with CRKP colonization had worse 100-day PFS ($P=0.0074$) than those

with CSKP colonization. Furthermore, 100-day relapse, NRM and OS were proved have no significant difference between two groups, with P values were 0.95, 0.07 and 0.079 respectively. We add these contents in the resubmitted manuscript and the detailed figures are shown in Supplementary Materials.

3) It is recommended that the author revise all baseline tables in the paper to three-line tables.

Response: Thanks for your reminder. In our resubmitted documents, we have revised all tables to three-line tables.

4) The caption (KP-mode, All, and None) of Figure 2 is unclear.

Response: Thanks for your careful checks. We are sorry for our carelessness. Based on your comments, we have revised Figure2 to make the caption clear.

5) There are many minor spelling errors in the text. The author needs to comprehensively check and revise them through the whole manuscript.

Response: We feel sorry for our carelessness. In our resubmitted manuscript, we have made the corrections. Thanks for your correction.

We tried our best to improve the manuscript and made some changes marked in red in resubmitted paper which will not influence the content and framework of the manuscript. We appreciate for Editors/Reviewers' work earnestly, and hope the correction will meet with approval. Once again, thank you very much for your comments and suggestions.

Re: Spectrum02039-23R1 (A clinical predictive model for pre-transplantation *Klebsiella pneumoniae* colonization and relevance for clinical outcomes in patients receiving allogeneic hematopoietic stem cell transplantation)

Dear Prof. Xiaojin Wu:

Your manuscript has been accepted, and I am forwarding it to the ASM production staff for publication. Your paper will first be checked to make sure all elements meet the technical requirements. ASM staff will contact you if anything needs to be revised before copyediting and production can begin. Otherwise, you will be notified when your proofs are ready to be viewed.

Sincerely,
Fei Chen
Editor
Microbiology Spectrum